Using laser micro-dissection and qRT-PCR to analyze cell type-specific gene expression in Norway spruce phloem

Nagy Nina E. 1 nina.nagy@skogoglandskap.no
Sikora Katarzyna 2
Krokene Paal 1
Hietala Ari M. 1
Solheim Halvor 1
Fossdal Carl Gunnar 1 carl.fossdal@skogoglandskap.no
1 Norwegian Forest and Landscape Institute , Ås , Norway
2 Forest Research Institute , Sękocin Stary, Raszyn , Poland
Huber Dezene
Electronic publication date: 2014 Apr 29
Publication date: 2014
Volume: 2
Electronic Location ID: e362
Received 2013 Dec 4; Accepted 2014 Apr 4
Copyright: © 2014 Nagy et al.
Copyright year: 2014
Copyright holder: Nagy et al.
License: This is an open access article distributed under the terms of the Creative Commons Attribution License, which permits unrestricted use, distribution, and reproduction in any medium, provided the original author and source are credited.
License URL: https://creativecommons.org/licenses/by/3.0/

Keywords: Ceratocystis polonica, Chitinase, Conifer defense, PAL, Pathogen infection, Phloem resistance response, Picea abies, Polyphenolic parenchyma cells, Ray parenchyma cells

Funding: Norwegian Forest and Landscape Institute EEA Financial Mechanism Norwegian Financial Mechanism This work was supported by the Norwegian Forest and Landscape Institute; Katarzyna Sikora was supported by the governments of Iceland, Liechtenstein and Norway through a grant from the EEA Financial Mechanism and the Norwegian Financial Mechanism under the Scholarship and Training Fund. The funders had no role in study design, data collection and analysis, decision to publish, or preparation of the manuscript.

==============================
The tangentially oriented polyphenolic parenchyma (PP) and radially organized ray parenchyma in the phloem are central in the defense of conifer stems against insects and pathogens. Laser micro-dissection enables examination of cell-specific defense responses. To examine induced defense responses in Norway spruce stems inoculated with the necrotrophic blue-stain fungus Ceratocystis polonica, RNA extracted from laser micro-dissected phloem parenchyma and vascular cambium was analyzed using real-time RT-PCR (qRT-PCR) to profile transcript levels of selected resistance marker genes. The monitored transcripts included three pathogenesis-related proteins (class IV chitinase (CHI4), defensin (SPI1), peroxidase (PX3), two terpene synthesis related proteins (DXPS and LAS), one ethylene biosynthesis related protein (ACS), and a phenylalanine ammonia-lyase (PAL). Three days following inoculation, four genes (CHI4, PAL, PX3, SPI1) were differentially induced in individual cell and tissue types, both close to the inoculation site (5 mm above) and, to a lesser degree, further away (10 mm above). These resistance marker genes were all highly induced in ray parenchyma, supporting the important role of the rays in spruce defense propagation. CHI4 and PAL were also induced in PP cells and in conducting secondary phloem tissues. Our data suggests that different cell types in the secondary phloem of Norway spruce have overlapping but not fully redundant roles in active host defense. Furthermore, the study demonstrates the usefulness of laser micro-dissection coupled with qRT-PCR to characterize gene expression in different cell types of conifer bark.

Introduction

Norway spruce (Picea abies (L.) Karst.), a dominant species in Europe’s boreal forests, is susceptible to the blue-stain fungus Ceratocystis polonica that is vectored into the stem by the spruce bark beetle Ips typographus L. During intermittent outbreaks, the beetle-fungus complex causes extensive tree mortality over large areas (Christiansen & Bakke, 1988; Wermelinger & Seifert, 1999). Norway spruce has an array of constitutive and inducible defense responses against insect-fungus attacks, including well-characterized structural and biochemical defense in the stem phloem (Franceschi et al., 2000; Franceschi et al., 2005). Among the cell types thought to be pivotal in bark defense are polyphenolic parenchyma (PP) cells and radial ray cells, which make up the major proportion of living cells of the phloem. PP cells are organized in annual rings of axially oriented parenchyma cells that form almost continuous sheets around the phloem circumference (Franceschi et al., 1998; Krokene, Nagy & Krekling, 2008). A characteristic feature of PP cells is the large vacuole that fills up most of the cell lumen and contains phenolic compounds (Franceschi et al., 1998). PP cells transport sugars to the surrounding parenchyma cells and store starch. The rays consist of parenchyma cells forming radial plates in the stem of conifers (Franceschi et al., 2005). Ray parenchyma store starch and form a living connection between the vascular cambium and the cork cambium, serving as a radial transport route for materials and signals in the bark.

Studies of the molecular basis of defense mechanisms in Norway spruce have shown up-regulation of resistance marker genes coding for chalcone and stilbene synthase, as well as pathogenesis-related (PR) proteins such as chitinase, defensin and peroxidases in infected bark (Fossdal et al., 2003; Fossdal et al., 2007; Fossdal et al., 2012; Nagy et al., 2004). Furthermore, the ethylene biosynthesis related synthase ACS (ACC synthase) and the lignin related peroxidase PX3 are involved in defense against pathogenic fungi in Norway spruce bark (Koutaniemi et al., 2007; Yaqoob et al., 2012; Deflorio et al., 2011). Immuno-cytochemistry has indicated that both rays and PP cells take part in production of secondary metabolites, since they contain abundant phenylalanine ammonia lyase (PAL), a key enzyme in phenol synthesis (Franceschi et al., 1998; Franceschi et al., 2000).

Until recently, all studies of defense related gene expression in Norway spruce and other conifers had to be done at the whole-tissue level. However, specific cell types such as rays and PP cells may have specialized roles in e.g., pathogen recognition and systemic defense signaling, and analysing whole tissues with a mosaic of different cell types does not resolve such cell- or tissue-specific processes. Laser micro-dissection (LMD) allows isolation of individual cell and tissue types and has provided new insight into the role of specific phloem cells in conifer defense responses (Li, Schneider & Gershenzon, 2007; Li et al., 2012; Abbott et al., 2010; Luchi et al., 2012). Combining LMD and sensitive chemical analyses Li, Schneider & Gershenzon (2007) showed that the lignified stone cells of Norway spruce bark also contain phenolic compounds, suggesting that these cells are involved in chemical as well as structural defense. More recently the same group showed that micro-dissected PP cells contain significantly higher concentrations of the stilbene glucoside astringin than neighboring sieve cells after infection with C. polonica (Li et al., 2012). LMD has also been used to characterize resin ducts and cambial tissue of white spruce phloem (Abbott et al., 2010) and to study expression of a constitutively expressed gene (α-tubulin) in micro-dissected Norway spruce phloem fragments consisting of multiple cell types (Luchi et al., 2012).

In this paper we use LMD to isolate ray and PP cells in Norway spruce phloem and analyze the differential gene expression of defense-related genes by quantitative Real Time RT-PCR (qRT-PCR). This increases the resolution of defense-related gene expression analysis in Norway spruce down to individual cell types and is the first investigation of gene expression in the two most important cell types in conifer bark defenses. We hypothesize that ray and PP cells play central roles in defense signalling and synthesis of defense compounds.

Materials and Methods

Inoculation and tissue sampling

Four 32-year-old ramets of a single Norway spruce clone (471), growing at the Hoxmark Experimental Farm of the Norwegian Forest and Landscape Institute in Ås, were used in this study. Clone 471 has strong, but not complete resistance to C. polonica infection, as determined by phloem lesion length following experimental inoculation (Nagy et al., 2005). It shows up-regulation of the phenylpropanoid pathway in the phloem following both fungal infection and mechanical wounding (Koutaniemi et al., 2007).

Two ramets (A and B) of clone 471 were inoculated with C. polonica (isolate no. NISK 93–208/115) on June 15, 2003, as described by Franceschi et al. (1998). Each ramet was inoculated at four sites in a circle around the stem between 1.5 and 2.0 m height. Tissue samples were collected 3, 7, 14 and 35 days after inoculation by removing a rectangular strip (2 × 10 cm) of bark including periderm, primary and secondary phloem, and cambium, with the inoculation site in the middle (Fig. 2A). At day 35, two control samples consisting of un-inoculated tissue were collected from ramet A at the same height, but 0.5 m away from the inoculation sites. All samples were frozen in liquid nitrogen immediately after harvesting and stored at −80 °C.

Figure 1 High resolution characteristics of Norway spruce phloem 0–14 days after inoculation with the necrotroph Ceratocystis polonica.

(A) Polyphenolic parenchyma cells (PPC) and ray cells (RC) in control tissue with turquois stained phenolics and unstained starch grains. (B) PPC and RC 3 days after infection, the time point at which cells and tissues were collected for laser micro-dissection and real-time qRT-PCR analysis. (C, D) Arrows indicate hyphae of C. polonica inside cells 7 and 14 days after inoculation. Bars, 50 µm. All cross-sections (1 µm thick) represent conducting secondary phloem sampled 5 mm above the inoculation site and embedded in acrylic resin.

Figure 2 Tissue regions and cell types in Norway spruce phloem selected for laser micro-dissection.

(A) Schematic outline of a bark sample showing the inoculation site and positions were tissue cross-sections were taken. (B) Overview of primary and secondary phloem in cross-section from the periderm to the cambium, with dissected regions and cell types dissected. 1: a 500 µm wide region of non-conducting primary phloem tissue adjacent to the outer bark. 2: a 500 µm wide region of non-conducting secondary phloem 1000 µm away from region 1. 3: a 500 µm wide region of conducting secondary phloem adjacent to the periderm. 4: cambium parenchyma cells. (C) Cross-section of cryo-embedded tissue showing polyphenolic parenchyma cells (PPC) and ray parenchyma cells (RC) in the secondary phloem selected for laser micro-dissection.

Cryo-sectioning

For cryo-sectioning and subsequent LMD we only included control samples (ramet A) and inoculated samples collected 3 days after inoculation (ramet A and B). Prior to cryo-sectioning, phloem cubes (5 × 5 × 5 mm) were cut from inoculated bark samples 5 and 10 mm above the inoculation site and embedded and frozen in Optimal Cutting Temperature embedding medium (Sakura Finetek USA, Inc., USA). Similar-sized phloem cubes were cut from the control samples and processed in the same way.

Transversal cryo-sections (20 µm thick) were cut from the upper part of each phloem cube using a cryo-microtome (Microm HM 560 MV, Microm International GmbH, Walldorf, Germany). Optimal sections were obtained at the temperature of −18 °C set for both specimen and knife. Ten sections, intended for LMD, were placed on nuclease and nucleic acid free PET-membrane frame slides (1.4 µm; Leica MicroDissect GmbH, Herborn, Germany) and stored immediately in falcon tubes on dry ice. From each phloem cube we prepared two slides with five cross-sections per slide. The slides were stored at −80 °C for a few days prior to laser micro-dissection. Additional cryo-sections for morphological characterization were cut and processed as described below for tissue examined by light microscopy.

Light microscopy

Thin transversal-sections were cut from phloem cubes embedded in LR White resin (TAAB Laboratories, Aldermason, Berkshire, UK) for routine observations of morphology and presence of fungal hyphae. Samples were processed for light microscopy according to Nagy et al. (2000). Briefly, pieces were fixated (in 2% paraformaldehyde and 1.25% glutaraldehyde in 50 mmol/L L-piperazine-N-N′-bis (2-ethanesulfonic) acid buffer (pH 7.2) for 12 h at room temperature), and dehydrated in an ethanol series (70-80-90-96-4 × 100%) before infiltration and polymerization (at 60 °C for 24 h) with L. R. White acrylic resin. Cross-sections (1 µm thick) were cut from all control samples and all inoculated samples collected 3–35 days after inoculation using an ultramicrotome for resin sections (Leica EM UC6; Leica Microsystems, Wetzlar, Germany). Both resin sections and the cryo-sections collected for morphological examination by light microscopy were dried onto superfrost® Plus glass slides (Menzel-Gläzer®; Thermo Scientific, Gerhard Menzel GmbH, Braunschweig, Germany) and stained with Stevenel’s blue (del Cerro, Cogen & del Cerro, 1980).

Laser micro-dissection (LMD)

LMD was performed as described by Abbott et al. (2010) with adaption to the subjected cell and tissue types. Cryo-sectioned phloem cross-sections were allowed to dry at room temperature for 5 min prior to micro-dissection with a LMD6000 Laser Micro-dissection Microscope (Leica Microsystems CMS GmbH, Wetzlar, Germany). For optimal dissection we used laser energy intensity between 85 and 100 and a cutting speed of 5. Different micro-dissected tissues were collected individually into the cap of nuclease free 0.5 ml PCR tubes (Axygen, Union City, CA, USA) containing 40 µl of lysis buffer. The tubes were then closed and centrifuged at low speed (3000 rpm) for 30 s to sediment the LMD samples, additional buffer was added to a final volume of 60 µl, and the tubes were placed on dry ice.

Both individual cell types and specific tissue regions were micro-dissected from the phloem cross-sections. Cell types selected for LMD at 20 × magnification were PP cells and ray parenchyma cells (RC), both occurring as clusters or rows of multiple cells in the secondary phloem. In addition, un-differentiated cambium parenchyma cells were dissected at 6.3× magnification. Tissue regions selected for LMD at 6.3× magnification were (1) primary phloem tissue near the periderm (old non-conducting phloem), (2) secondary phloem tissue ∼1000 µm centripetal to region (1) (non-conducting secondary phloem), and (3) secondary phloem adjacent to the cambium (conducting secondary phloem) (Fig. 2B). All tissue regions measured approximately 500 × 2000 µm (radial × tangential dimension). The total cross-sectional area dissected from the 10 cross-sections per phloem cube was approximately 400 000 µm2 for individual cell types and 3 000 000 µm2 for tissue regions. When different cell types and tissue regions were micro-dissected from the same cross-section, dissection of one was always completed before starting the next to avoid cross-contamination.

RNA extraction and boosting

RNA was extracted independently from each tissue region and cell type, following the protocol for the RNAqueous-Micro RNA Kit (Ambion, Inc., Austin, TX, USA). Briefly, tissues and cells were collected in RNA lysis buffer. Prior to cell lysis at 42 °C, buffer was added to give a total volume of 100 µl lysate for a silica column-based purification with elution of total RNA, using 12 µl elution buffer heated to 95 °C. DNase treatment was performed following the manufacturer’s protocol. Due to differences in mRNA yield between tissue regions (consisting of a mixture of different cell types) and pools of single cells, RNA extracts from single cells were boosted using the MessageBOOSTER cDNA Synthesis Kit for qPCR (Epicentre Biotechnologies, Madison, WI, USA) to obtain sufficient quantities of cDNA for qRT-PCR analyses (Yakovlev et al., 2006).

Quantitative real time RT-PCR (qRT-PCR) analysis

In order to identify tissue- and cell type-specific molecular defense responses, we analyzed the expression of several transcripts using quantitative real-time reverse transcription PCR (qRT-PCR). The qRT-PCR reactions were performed in single-plex conditions in a 96-well plate sealed with a plastic film, using a 7500 Real Time PCR System (Applied Biosystems, Foster City, CA, USA) with a reaction mixture consisting of 1× SYBR Green PCR Master Mix (Applied Biosystems, Warrington, UK), 120 nM of each primer, and 5 µl of cDNA. The PCR program was 2 min at 50 °C, 10 min at 95 °C followed by 40 cycles of 15 s at 95 °C and 1 min at 60 °C.

Absolute transcript quantification was performed using the 7500-system’s SDS software (Applied Biosystems, Foster City, CA, USA). The expression level of each target gene was normalized to the transcript level of the endogenous control actin (PaAct), both in infected and control samples. PaAct was used as the endogenous control reference since it has been shown to have the best stability index value among several tested genes, including α-Tubulin (PaαTub), glyceraldehyde-3-phosphate dehydrogenase (PaGAPDH) and polyubiquitin (PaUbq) (Yakovlev et al., 2006). Samples with low RNA yield (cycle threshold (Ct) value for actin above 35) were excluded from candidate gene transcript profiling. The targeted gene transcript levels were profiled by gene specific primers described and verified in previous studies (Table 1).

Table 1 Primer sequences used for real time qRT-PCR analysis of micro-dissected Norway spruce phloem tissues and cells.

Gene	Gene name and reference	GenBank Accs.*	Primer sequences (forward/reverse, 5′–3′)	
Act	Actin (Yakovlev et al., 2006)	AAF03692	TGAGCTCCCTGATGGGCAGGTGA/TGGATACCAGCAGCTTCCATCCCAAT	
CHI4	Chitinase (Yaqoob et al., 2012; Fossdal et al., 2007)	AY544780	GCGAGGGCAAGGGATTCTAC/GTGGTGCCAAATCCAGAAA	
SPI1	Defensin (Fossdal et al., 2007)	X91487	TGTGGCCAACAGAAAGTGCTA/CCAGTGAAGATCACAGTAGTAGGATTAGG	
PX3	Peroxidase (Koutaniemi et al., 2007; Yaqoob et al., 2012)	AJ566203	ATGGTGGCGCTGTCAATTC/TGCTGTAGAACGTCCAAGAAAGAC	
PAL	Phenylalanine ammonialyase (Deflorio et al., 2011)	AY639588	CAGCCCTCTGCCCAACAG/AGCTGGGTTCTCACGAATTCA	
DXPS	1-deoxyxyulose-5-phosphate
synthase (Abbott et al., 2010)	EF688333	AGAAACTCCCTGTGAGATTTGCCCTT/CAACAGTAACTGATATGCCCTGCTGAG	
LAS	Levopimaradiene diterpene
synthase (Abbott et al., 2010)	AY473621	GGACGATCTCAAGTTGTTTTCCGATTC/TGAGAACCACTGTTCCCAGCGC	
ACS	1-aminocyclopropane-1-carboxylate synthase
(Yaqoob et al., 2012)	BT108790	CAAGCAGAATCCCTATGATGCCGAAA/TCTGGATGAGACTTGAGCCAACCTTC	
TIF	Translation initiation factor
(Abbott et al., 2010)	AY961930	CATCCGCAAGAACGGCTACATC/GTAACATGAGGGACATCGCAG	
Notes.

* References denote related studies where these gene transcripts were used.

Results

Phloem colonization by C. polonica

No hyphae were observed in phloem 5 or 10 mm above the inoculation site 3 days after inoculation and the tissue showed no anatomical changes compared to control tissue (Figs. 1A and 1B). However, by day 7 and 14, C. polonica hyphae had extended 5 to 10 mm away from the inoculation site, mostly growing within the lumen of ray cells and PP cells (Figs. 1C and 1D). By day 35, hyphae were observed growing into and through large phenolic aggregations present within these cells.

Identification of cells for laser micro-dissection

Cryo-sectioned tissues showed cell morphology that was in an adequate state of preservation. Before laser micro-dissection we were able to identify ray and PP cells based on morphological features such as cell shape, orientation, cellular tissue architecture and cytoplasmic content. In phloem cross-sections PP cells can be identified by their rounded shape, their occurrence in multiple axial rows separated by 5–7 layers of empty sieve cells, and the presence of polyphenolic globules and starch grains in their cell lumen. The ray parenchyma cells have a characteristic radially elongated shape, form chain-like rows radiating through the bark and extending into the xylem, and contain polyphenolic aggregates and starch grains that appear irregularly in small amounts (Figs. 2B and 2C).

Transcript levels in control material

TIF was constitutively expressed at levels similar to our endogenous reference actin in both whole tissue regions and individual cell types. Transcript levels of CHI4, PAL, SPI1 and PX3, were detectable in some un-inoculated tissue, but at a low to very low level (Table 2). This was also the case for DXP, LAS and ACS (data not shown). The low constitutive expression level of these genes was further confirmed in additional control samples that were collected from two un-inoculated ramets of clone 471 in July 2010 (see Supplemental Information). The average fold-difference in transcript levels between the 2003 and 2010 controls was <0.25 for tissue types and 0.43 for cell types (average for all three tissue/cell types and all the eight target gene transcripts studied).

Table 2 Expression profiles of five genes in different tissue regions and cell types of Norway spruce phloem, after inoculation with Ceratocystis polonica and in control.

Gene expression was determined in sections taken 5 and 10 mm above the inoculation site in ramet A and B of clone 471. Data are presented as relative transcript abundance normalized to actin expression. Dash (—) indicates that the sample was not subjected to target gene profiling due to low RNA yield (cycle threshold value for actin above 35).

Gene	Tissue and cells	Infected	Control	
		Ramet A
5 mm d3	Ramet A
10 mm d3	Ramet B
5 mm d3	Ramet B
10 mm d3	Ramet A
site 1 d35	Ramet A
site 2 d35	
CHI4	Primary phloem	3.03	—	0.86	1.38	0.00	0.01	
	Sec. phloem conducting	19.52	1.32	6.78	1.74	0.00	0.04	
	Sec. phloem non-conducting	49.84	1.76	2.45	0.60	0.00	0.00	
	Cambium	5.21	1.03	0.41	—	0.20	0.02	
	Ray cells	51.66	3.87	4.36	0.49	0.00	0.00	
	PP cells	29.79	3.64	2.35	0.30	0.00	3.71	
PAL	Primary phloem	3.72	—	2.23	1.68	0.02	0.05	
	Sec. phloem conducting	2.44	1.89	4.34	0.81	0.14	0.14	
	Sec. phloem non-conducting	7.42	2.07	3.20	0.87	0.03	0.02	
	Cambium	6.09	1.09	2.62	—	0.08	0.05	
	Ray cells	8.21	4.85	4.27	1.89	0.00	0.00	
	PP cells	3.65	6.42	1.60	2.33	0.00	0.38	
SPI1	Primary phloem	0.81	—	0.00	0.00	0.00	0.00	
	Sec. phloem conducting	0.37	2.54	0.00	0.80	1.32	0.42	
	Sec. phloem non-conducting	0.41	2.18	0.53	0.11	2.48	0.31	
	Cambium	0.36	4.31	4.13	—	0.00	0.00	
	Ray cells	3.96	5.74	2.29	5.35	0.00	0.00	
	PP cells	0.46	0.47	0.34	1.34	0.00	0.00	
PX3	Primary phloem	0.07	—	0.10	0.05	0.00	0.01	
	Sec. phloem conducting	0.24	0.12	1.22	0.83	0.00	0.00	
	Sec. phloem non-conducting	0.33	0.39	0.53	0.16	0.00	0.00	
	Cambium	0.37	0.43	13.73	—	0.94	0.00	
	Ray cells	1.90	0.38	11.07	1.03	0.00	0.00	
	PP cells	0.06	0.00	0.18	0.11	0.00	0.00	
TIF	Primary phloem	1.59	—	0.90	0.71	0.76	0.48	
	Sec. phloem conducting	0.99	1.17	0.81	0.59	3.11	0.70	
	Sec. phloem non-conducting	2.18	1.32	0.68	0.62	0.94	0.60	
	Cambium	1.28	1.10	2.59	—	2.84	1.13	
	Ray cells	3.28	1.75	2.65	1.82	0.69	1.03	
	PP cells	2.20	0.83	0.67	1.55	0.00	0.00	
Notes.

Ramet trees of Norway spruce clone number 471

mm distance from inoculation site

d day

Transcript levels in different phloem regions of infected bark

The analysed phloem tissue regions consisted of several cell types including living parenchyma cells (ray and PP cells), dead sieve cells, and cambium parenchyma. Three days after inoculation, CHI4, PAL and PX3 showed increased transcript levels in all three tissue regions examined (primary phloem near the bark surface, non-conducting and conducting secondary phloem), but generally less so in the primary phloem (Table 2). Upregulation level of CHI4, PAL and PX3 showed some ramet-specific variation but commonly dropped from 5 to 10 mm above the inoculation site. The terpene related genes DXPS and LAS (data not shown), the ethylene biosynthesis related ACS (data not shown), as well as SPI3 and TIF showed no consistent induction in phloem 3 days after inoculation, neither close to (5 mm) or further away (10 mm) from the inoculation site (Table 2).

Transcripts levels in different cell types of infected bark

As with tissues, the laser micro-dissected cell types also displayed some ramet-specific variation but with clear induction of CHI4, PAL and PX3 transcripts 3 days after inoculation. Maximum transcript levels of these genes were generally recorded in the area adjacent to the site of inoculation (Table 2). The highest transcript levels of CHI4, PAL and PX3 were usually observed in ray cells. Regarding PAL, the three targeted cell types showed a relatively similar induction level. Cambium displayed clearly lower induction of CHI4 than ray and PP cells, whereas PP cells showed the lowest induction level of PX3. No clear induction compared to cells from un-inoculated control tissue was observed for DXPS, LAS, ACS (data not shown), SPI1 or the constitutively expressed TIF (Table 2).

Discussion

This study demonstrated the usefulness of the laser micro-dissection technique coupled with qRT-PCR for deciphering cell type-specific induced defense responses in conifer phloem. The LMD procedure followed in this paper did not include chemical fixation and histological stains, but still allowed identification of the various cell types based on morphological characteristics. Conventional cytological analyses rely on tissue treatment with fixatives and stains, which are likely to impact subsequent extraction and quality of RNA. When the constitutive bark defense of conifers is compromised, induced host responses are launched in the neighboring tissue in order to kill or compartmentalize the invader. We focused on short-term molecular defense responses occuring within 3 days after fungal inoculation in tissues that locate 5–10 mm away from the site of bark wounding and infection. At this early stage of infection these adjacent tissues are not visibly damaged and compartmentalization associated anatomical and chemical changes have not yet been completed. The use of phloem cross-sections enabled us to evaluate the fungal colonization status of the whole phloem region in a single cut extending from the outer bark into the cambium, and also allowed dissection of different tissue regions and cell types from the same section. The tangential sectioning procedure used by Abbott et al. (2010) to study resin ducts would have required a larger number of sections to be cut in our case.

In comparison to infected bark, the transcripts of CHI4, PAL, and PX3 were either absent or at a very low level in all analyzed tissue and cell types of control bark (Table 2). The contrasting transcript levels of these genes between infected and control bark are consistent with a role in induced bark defense.

The class IV chitinase CHI4 was up-regulated close to the inoculation site particularly in ray and PP cells. CHI4 has been demonstrated to have antifungal effects (Ubhayasekera et al., 2009) and has been indicated to mediate programmed cell death (PCD) during embryogenesis in Norway spruce (Wiweger et al., 2003). Mediation of PCD by CHI4 presumably occurs through action on endogenous arabinogalactan proteins or lipo-chitooligosaccharides (Wiweger et al., 2003), conceptually releasing oligosaccharides with signaling properties (Fossdal et al., 2006). CHI4 was much more highly induced in ray and PP cells close to the inoculation site than further away. This pattern may be caused by PCD that precedes formation of a ligno-suberized boundary zone (LSZ), a process in which PP cells are instrumental (Franceschi et al., 1998; Franceschi et al., 2000; Franceschi et al., 2005).

Phenylalanine ammonia lyase (PAL), the basal enzyme in the phenylpropanoid pathway, is upstream of both lignin and polyphenol synthesis. PAL was also up-regulated in both ray and PP cells, although to much lower levels than CHI4. In all tissues PAL was most strongly up-regulated close to the inoculation site and this may be due to increased monolignol production and lignification of the LSZ that is forming to contain the infection. However, clarification of downstream events in cells with PAL induction will require transcript profiling of genes specific to lignin biosynthesis and phytoalexin formation. The phenolic content of PP cells can be rapidly activated and modified following infection, since the PP cells have ample energy supplies in the form of stored starch and lipids (e.g., Krekling et al., 2000). Activation or swelling of PP cells following infection results in a four-fold increase in their volume and changes in the appearance of their phenolic content (Franceschi et al., 1998; Franceschi et al., 2000; Nagy et al., 2000; Nagy et al., 2004). The swelling leads to extensive compression of the surrounding sieve cells, transforming the induced phloem into dense blocks of cell walls separated by layers of swollen PP cells. This combined cell wall/PP cell barrier appears to be reinforced by phenolics that are released from the induced PP cells and deposited in the surrounding sieve cell walls (Franceschi et al., 2000). In Sitka spruce cell-wall bound phenols (but not lignin) have been shown to increase rapidly in the bark 10 mm from inoculation sites following inoculation with H. annosum (Deflorio et al., 2011).

Up-regulation of the peroxidase PX3 was very pronounced in the cambium and in ray cells close to the site of wounding and inoculation. Peroxidases appear to play a role in cell wall formation by providing the radical-generating capability for coupling individual phenolic monomers into complex lignin polymers (e.g., Ralph et al., 2004). In Norway spruce PX3 is believed to be involved in oxidative processes leading to increased cell wall lignification (Koutaniemi et al., 2007). This is supported by earlier experiments showing increased levels of lignification in the bark following wounding and inoculation with fungus (Fossdal, Sharma & Lönneborg, 2001; Deflorio et al., 2011). Likewise, Koutaniemi et al. (2007) found highly induced levels of PX3 in compression wood and bark of Norway spruce trees inoculated with the decay fungus Heterobasidion parviporum but not in normal developing xylem. The low transcript levels of PX3 in PP cells are noteworthy and might be due to the role of phenolics released from PP cells in reinforcing the cell walls of the surrounding sieve cells. An alternative possibility is that a major part of the phenolics synthesized by PP cells upon bark compromise are soluble rather than cell-wall bound. The translation initiation factor TIF was used as an endogenous qRT-PCR reference transcript in a related study of bark defenses in white spruce by Abbott et al. (2010). This fits well with the constitutive expression levels observed in our study, where TIF had similar expression levels as our endogenous reference actin. The expression level of genes involved in terpenoid resin synthesis (DXPS, LAS, ACS) showed no clear induction in our study. For example LAS (levopimaradiene abietadiene synthase), a major enzyme in diterpene biosynthesis, was almost undetectable in our samples. This suggests that our targeted cell types are not involved in resin biosynthesis and accords well with earlier findings in white spruce and Sitka spruce, where LAS activity was found to be localized to cortical resin ducts and to be absent from other tissues (Abbott et al., 2010; Zulak et al., 2010; Zulak & Bohlmann, 2010).

The use of a single Norway spruce genotype in this study brings into question the general applicability of our results. Our findings are strengthened by the consistent differences observed between control and infected bark, between tissue and cell types, with distance from the inoculation site, and between different gene transcripts. Further, our data are consistent with previous studies describing the regulation of these gene transcripts in bulk tissues of other Norway spruce genotypes following wounding and pathogen inoculation. For example, CHI4 was the most highly up-regulated transcript in this study (Table 2), as has been shown in numerous earlier studies (Fossdal et al., 2006; Fossdal et al., 2007; Fossdal et al., 2012; Yaqoob et al., 2012). Our previous study suggested that resistant and susceptible Norway spruce clones differ in the rapidity of PaCHI4-related signal perception or transduction in the challenged inner bark (Hietala et al., 2004); it remains to be examined whether resistant and susceptible clones differ in the rate and level of reprogramming of all cell types upon wounding/pathogen challenge.

Conclusions

Laser micro-dissection greatly increases the resolution of mRNA analyses by allowing gene expression profiling of specific cell types. We applied this method to analyze gene expression in the key cell types involved in defense of Norway spruce phloem, namely polyphenolic parenchyma (PP) and ray parenchyma cells. Laser micro-dissection is particularly applicable in combination with qRT-PCR when preserving chemicals are avoided. While the low number of replicates limits the biological conclusions that can be drawn from this study, we have demonstrated that the induced expression profiles of particularly CHI4 and PX3 differed between vascular cambium, PP cells and ray parenchyma, suggesting that these cell types have overlapping but not fully redundant roles in active defense of Norway spruce phloem.

Supplemental information

Supplemental Information 1 Expression profiles of five genes in different tissue regions and cell types of Norway spruce phloem, after inoculation with Ceratocystis polonica and in controls

Gene expression was determined in sections taken 5 and 10 mm above the inoculation site in ramet A and B of clone 471. Control samples are from unharmed tissues of ramet A, C and D of clone 471. Data are presented as relative transcript abundance normalized to actin expression. Dash (—), indicates that the sample was not subjected to target gene profiling due to low RNA yield (cycle threshold value for actin above 35).

Click here for additional data file.

Supplemental Information 2 Expression profiles of five genes in different tissue regions and cell types of Norway spruce phloem, after inoculation with Ceratocystis polonica and in control

Gene expression was determined in sections taken 5 and 10 mm above the inoculation site in ramet A and B of clone 471. Data are presented as relative transcript abundance normalized to actin expression. Dash (—), indicates that the sample was not subjected to target gene profiling due to low RNA yield (cycle threshold value for actin above 35).

Click here for additional data file.

Additional Information and Declarations

Competing Interests

Author Contributions

Nina E. Nagy, Paal Krokene, Ari M. Hietala, Halvor Solheim and Carl Gunnar Fossdal are employees of the Norwegian Forest and Landscape Institute; Katarzyna Sikora is an employee of the Forest Research Institute in Poland. The authors of this manuscript have no financial, professional, personal, or other competing interests to declare, which would have otherwise caused bias regarding this work.

Nina E. Nagy conceived and designed the experiments, performed the experiments, analyzed the data, contributed reagents/materials/analysis tools, wrote the paper, prepared figures and/or tables, reviewed drafts of the paper.

Katarzyna Sikora performed the experiments, analyzed the data, contributed reagents/materials/analysis tools, wrote the paper, reviewed drafts of the paper.

Paal Krokene conceived and designed the experiments, performed the experiments, analyzed the data, wrote the paper, reviewed drafts of the paper.

Ari M. Hietala conceived and designed the experiments, analyzed the data, wrote the paper, reviewed drafts of the paper.

Halvor Solheim conceived and designed the experiments, contributed reagents/materials/analysis tools, wrote the paper, reviewed drafts of the paper.

Carl Gunnar Fossdal conceived and designed the experiments, performed the experiments, analyzed the data, contributed reagents/materials/analysis tools, wrote the paper, reviewed drafts of the paper.

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
