# Peer review of "Using laser micro-dissection and qRT-PCR to analyze cell type-specific gene expression in Norway spruce phloem"

_PeerJ, doi:10.7717/peerj.362_

## Round 0.1 · original submission · Major Revisions

I would like to thank both reviewers for their thoughtful comments. One reviewer recommends acceptance with minor revisions, the other recommends major revisions. I am also recommending major revisions to this MS.

Specifically:

-I agree with one reviewer that the number of replicates is very low. This is not fatal, of course. Many physiological studies "cope" with few replicates due to the difficulty and/or expense of obtaining higher numbers of replicates. However, this does have ramifications on the overall general applicability of the findings. Can the authors increase the replicates, or at least discuss this low number of replicates in the context of general application of the results? Note that the reviewer acknowledges the difficulty in obtaining workable levels of RNA in this method. And that is fair. But, again, limitations should then be discussed. Note also that at 10mm and for two tissue types (primary phloem and cambium) there was only one replicate reported. This is very low and, as figure 3 makes clear, makes it impossible to assign a standard error. This has statistical ramifications related to my comments on statistical methods below.

-The same reviewer also points out that there is a lack of clarity on which ramets were used for which treatment/control group.

-It was also mentioned, and I agree, that there is some potential difficulty with the fact that treatment and control groups were collected seven years apart from each other. The experimental design as currently described does not allow for testing of variation due to time of collection. It is quite feasible that annual environmental conditions (or perhaps insect or pathogen infestation/infection in the intervening seven years) is the actual reason for differential gene expression.

-The authors cite a paper (Deflorio et al. 2011) in their M&M section that seems to indicate that they used the identical methods to analyze their data. Here is a copy-and-paste direct quote of the relevant portion of the text of the Deflorio et al. (2011) paper:

"Gene expression values appeared skewed and so were log-transformed and analysed for each tissue type separately using analysis of variance. Individual ‘tree’ was a random effect, whereas ‘clone’, ‘untreated’ (untreated vs. wounded/inoculated), ‘fungus’ (wounded only vs. wounded and inoculated), and ‘location’ (A, B), and their interaction terms, were fixed effect terms. In addition, an analysis of the combined data was carried out, with ‘tissue’ type and its interactions as an additional fixed effect. Where the analysis of variance indicated significant effects, the nature of significant differences was explored using a post-hoc t-test.

Phenolic and lignin data were also log-transformed when their distribution appeared skew and subsequently analysed for each tissue type using analysis of variance. Individual ‘tree’ was a random effect and ‘clone’ and ‘treatment’ (untreated vs. wounded and inoculated), and their interaction terms, were fixed effect terms. All statistical computations were done using R [25]."

…it is not clear in the current MS if those are precisely the same statistical methods employed. My guess is that there were some variations, since the experimental parameters were different. It also seems possible that the data had different characteristics (e.g., skew) requiring different treatment.

As such, the authors should explicitly state their statistical methods here, perhaps maintaining the reference to Deflorio et al. but discussing how their analyses differed. The statistical test details should also be present in the associated figure (figure 3) and figure heading.

Both reviewers also provide a number of other very useful comments that the authors should address.

As one reviewer notes, "The authors are very experienced in this sort of work…" And this gives me confidence that the authors can respond to these questions and criticisms.

Thank you for your submission to PeerJ, and I look forward to seeing the revised manuscript.

Reviewer 1 ·

Basic reporting

No Comments.

Experimental design

Inoculation and Tissue Sampling - In line 73 the authors cite June 15 2003 as the initial date for the inoculation of Norway spruce with the pathogen. Controls were collected in July of 2010 in line 79. Why the discrepancy between collections? Can the authors add an additional line of justification citing why these are comparable tissue types to compare? Only 2 biological replicates were used in this study for each treatment combination. It is unclear whether LMD sections from ramet A and B were combined or only one ramet was used for LMD, while the other was used for light microscopy. The same goes for control samples (ramets C and D). The methods should be more detailed so the reader can understand where each ramet is used.

Cryosectioning – It is unclear if the authors are using a method for cryosectioning that has been previously published (i.e. Abbott et al. 2010) or are employing their own method. Also, please cite section thickness used and captured from the cryostat.

Light Microscopy – What samples were used for light microscopy analyses? Were the samples fixed and dehydrated before LR White resin embedding? If so, please state the fixation medium and methods used.

LMD – Were sections dehydrated in EtOH as in Abbott et al. 2010 or were they cut fresh and allowed to dry at RT? If so, was RNA integrity analyzed (RIN) to see if RNA quality was sufficient for downstream applications? How long were collection tubes open? Did lysis buffer dry out during collection? Please state how this method differs from or is different from Abbott et al. 2010.

RNA Extraction and Boosting – Were all tissue types boosted and treated equally?

Validity of the findings

Based on the concerns listed under the experimental design, I have reservations about the overall findings of the study. Due to the limited number of biological replicates and only one genotype, it is difficult to judge if the results from this study are representative of a broader population. Also, due to differences in collection times of control vs. inoculated samples and the use of only 2 biological reps, it is impossible to know what type of variation may be present. I say this, while also taking into consideration the difficulty of isolating enough RNA using LMD. However, greater care may have been taken to standardize the sampling process, and the addition of an extra biological rep that was then pooled may have been more representative and allowed for more confidence with the results.

Additional comments

The authors present a nice study that adds unique information to the literature relating to cell specific responses in conifers and defense. However, some clarification and reasoning is needed before this can be accepted for publication as outlined above.

·

Basic reporting

This manuscript reports to use of laser microdissection to determine gene expression in specific cell types during the response of Norway spruce bark to inoculation with the virueltn pathogen Ceratocystis polonica. It is very well written and based on excellent scientific protocols and data interpretation.

Experimental design

The authors are very experienced in this sort of work, and the experimental design refelcts this expertise.
I have a few comments on the methods, though they are not particularly significant in terms of the work done.
1. What is cork bark? I think they mean phellem tissues? Better to use the technically correct wording than the rathe vague 'cork bark'.
2. Line 97: how do observing fungal infection and presence of fungal hyphae actually differ?
3. Line 119: 'cork bark again' The phloem they refer to here is actually secondary or redundant phloem, as only the phloem closest to the vascular cambium is functioning in transport.

Validity of the findings

The results are extremely interesting, and relevant to many host trees, plus many pathogens; possibly insect pests too. Further development of laser dissection, coupled with microscopy, will enable great advances in our understanding of the manner in which trees respond to pathogens, when both chemical and morphological responses can be linked directly to gene regulation. I have no specific comments on how the results (and discussion) are presented, except that I would prefer not to see references used in the Results (e.g. line 187): if a sentence requires a references, then it should probably be in the discussion section.

Additional comments

ALthough generally well written, the English is rather 'loose' in palces and requires some attention. For example , in line 123, use 'approximately', not 'about'! Also, use the past tense throughout - see the beginning of the discussion: 'This work demonstrated the utility of the laser micro-...'.
There's a curious '(,)' in line 267.
I do not intend to state all the language revisions necessary, but the authors may wish to ask a native speaker of English to check the language for them - I cannot see it would take very long.

---

## Round 0.2 · Major Revisions

Thanks to the authors for their revisions and to one of the reviewers for re-reviewing the paper.

The reviewer suggests "major revisions." Reading the suggestions, it seems to me that while the suggested revisions are reasonably major, they do sit substantially closer to the minor side of the spectrum as they mainly entail changing the analysis and presentation/interpretation of the data. In addition, the reviewer does mention substantial improvement.

In terms of one of the reviewer's suggestions, namely:

"The difference in sampling between ramets A and B vs. C and D is very confusing and does not add to the story. While the results do serve as a confirmation, they also distract from the overall story. I think the article could be streamlined if these data were removed from the study. It could be mentioned in the results that further confirmation of controls were carried out using samples collected in 2010, etc. However, incorporating these data with the earlier collected samples muddles the overall message and detracts from the story."

...here are my thoughts. Assuming the authors agree to remove C and D data from the main MS, it would still be useful to mention the confirmation from 2010 samples and to point to a supplementary file with the data and an appropriate figure or data. That way these important data will not be lost, and anyone who wants to judge the confirmation for themselves can do so.

I would like to encourage the authors to adjust this MS and/or sufficiently rebut the reviewer's thoughts, and I'd again like to thank everyone for taking part in this important process of scientific communication.

·

Basic reporting

No comments.

Experimental design

Inoculation and Tissue Sampling - The authors have improved this section greatly. My major concern, now that I better understand the material under investigation, is that the use of the controls (ramets C and D) seem out of place. I don't believe that these samples are necessary. It adds confusion and raises questions regarding the experimental design. I believe the story would be much cleaner if these samples were removed from the study and the focus be shifted to ramets A and B. The difference in sampling between ramets A and B vs. C and D is very confusing and does not add to the story. While the results do serve as a confirmation, they also distract from the overall story. I think the article could be streamlined if these data were removed from the study. It could be mentioned in the results that further confirmation of controls were carried out using samples collected in 2010, etc. However, incorporating these data with the earlier collected samples muddles the overall message and detracts from the story.

Validity of the findings

I find it difficult to understand and interpret the authors data, particularly figure 3, in its current format. A major limitation of the study is the low number of biological replicates which makes interpretation of the data difficult. However, this limitation is decreased because of the authors use of LMD. The difficulty associated with RNA isolation from LMD samples is understood, and should be made clear in the text for the reader as well. I would suggest that instead of attempting to force the data and make it behave as if there were biological replicates/technical replicates (figure 3) that the authors present the data on a replicate by replicate basis. The numbers could be presented just as easily in a table format, with each replicate being a separate row, and individual expression values present. Letting the individual replicates stand on their own will allow the reader to better understand the tedious and time consuming nature of LMD experiments. I would refer the authors to Li et al. 2007 which will help in clarification of this point.

Presenting data as in figure 3 with an n=2 and associated error bars does not make sense and should be removed, as a minimum of 3 reps is necessary for such an analysis. and n=3 is very confound considering it includes a replicate from 2003 and 2 from 2010. Breaking each rep down 1 by 1 will improve on this problem.

Additional comments

The study in its current format will need some refinement before being acceptable for publication in PeerJ. The presentation of results could be improved and in so doing (as mentioned in the previous section) will also help to emphasize the limitations associated with LMD studies. By changing the way the data are presented, it would also help to clarify the issue with the 2010 sampling date. The data would no longer be linked together and could stand on their own.

---

## Round 0.3 · accepted · Accept

Thank you to the co-authors and to the reviewers who have worked hard on this contribution. At this point, the authors have dealt with the various points mentioned by the reviewers and by me and the paper is acceptable for publication in PeerJ.